# Attention-Guided Black-box Adversarial Attacks with Large-Scale Multiobjective Evolutionary Optimization

**Jie Wang** [1]  **Zhaoxia Yin** [1]  **Jing Jiang** [2]  **Yang Du** [1]

## Abstract

Recent black-box adversarial attacks may struggle to balance their attack ability and visual quality of the generated adversarial examples (AEs) in tackling high-resolution images. In this paper, We propose an attention-guided black-box adversarial attack based on the large-scale multiobjective evolutionary optimization, termed as LMOA. By considering the spatial semantic information of images, we firstly take advantage of the attention map to determine the perturbed pixels. Then, a large-scale multiobjective evolutionary algorithm is employed to traverse the reduced pixels in the salient region. Extensive experimental results have verified the effectiveness of the proposed LMOA on the ImageNet dataset.

## 1. Introduction

In the past decade, a series of studies have shown that DNNs are vulnerable to adversarial examples (AEs) by imposing some designed perturbations to original images (Szegedy et al., 2013; Goodfellow et al., 2014b; Carlini & Wagner, 2017). These perturbations are imperceptible to human beings but can easily fool DNNs, which raises invisible threats to the vision-based automatic decision (Kurakin et al., 2016; Yin et al., 2020). Consequently, the robustness of DNNs encounters great challenges, and the issue of AEs has received considerable attention (Zhang & Li, 2019).

Szegedy *et al.* (2013)first pointed out the vulnerability of DNNs and proposed the definition of adversarial attacks. They also demonstrated that the AEs for one network could fool another, even DNNs were trained on different datasets. Then, a considerable amount of researches on adversarial

---

[1]Anhui Provincial Key Laboratory of Multimodal Cognitive Computation,School of Computer Science and Technology, Anhui University, Hefei, China. [2]School of Computer Science and Communication Engineering, Jiangsu University, Zhenjiang, China.. Correspondence to: Zhaoxia Yin <yinzhaoxia@ahu.edu.cn>.

*Accepted by the ICML 2021 workshop on A Blessing in Disguise: The Prospects and Perils of Adversarial Machine Learning.* Copyright 2021 by the author(s).

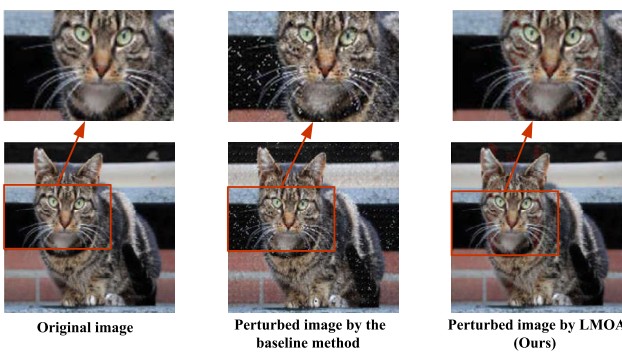

Figure 1. The original image and its corresponding AEs generated by the baseline (Suzuki et al., 2019) and LMOA, respectively.

attacks has been studied. These attacks are designed to fool the target DNN by adding a small perturbation $X$ to the original image $I : \mathcal{C}(I + X) \neq \mathcal{C}(I)$, where $\mathcal{C}(\cdot)$ is a $m$-class classifier that receives $n$-dimensional input and gives $m$-dimensional output. AEs can be easily generated by using internal information of the target DNN, *e.g.,* the gradient of the loss function of the original image (Goodfellow et al., 2014a; Kurakin et al., 2016; Madry et al., 2018; Dong et al., 2018). These attacks, called white-box attacks, are essentially an exploration of the robustness of DNNs. Other than the white-box attacks, researchers have shown an increased interest in black-box attacks. To be specific, the attacker can only obtain the output of the target DNN without accessing its structures and parameters. Since the structural prior knowledge of DNNs is usually unavailable, the works on black-box attacks are more practical than that of white-box ones in real cases. Therefore, numerous attempts have been made to realize black-box attacks(Bhambri et al., 2019), such as the gradient estimation-based (Tu et al., 2019), local search-based (Chen et al., 2019), or transferability of AEs-based (Dong et al., 2019) attacks.

To date, several works suggest that there is more than one objective should be taken into consideration in attacking, *e.g.,* minimizing the confidence probability of the true label and the perturbation intensity of the changed image simultaneously (Liu et al., 2020; Suzuki et al., 2019). They expected that the candidate AEs would mislead the target

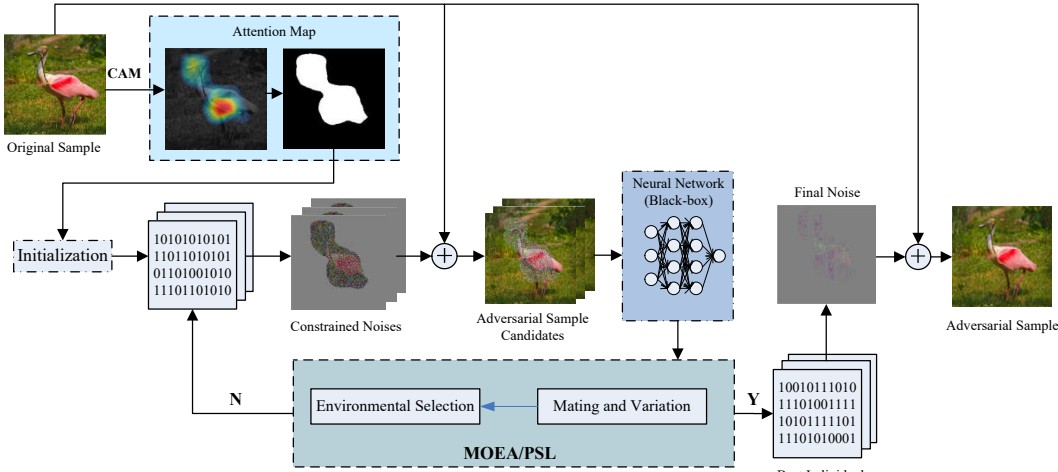

*Figure 2.* The general framework of the proposed LMOA.

DNN as possible while exhibiting similar visual features with the original image. Unfortunately, these two objectives are somewhat conflicting with each other. On the one hand, the great perturbations significantly influence the classification result of DNNs, but the generated AEs could be easily detected by the human vision, as they lose the majority of features of the original image. On the other hand, a slight change that is hardly observed by both human beings and computers is not enough to mislead DNNs. Most of the previous methods cannot balance these two objectives and thereby probably miss the optimal trade-off perturbation. The issue stimulates the efforts on evolutionary algorithm-based, especially multiobjective evolutionary algorithm-based black-box adversarial attacks. Moreover, since the traditional evolutionary algorithm are not competitive on handling the optimization problems with large-scale decision variables (Yang et al., 2008), the existing works following this line may not achieve satisfactory performance on the high-resolution images(Su et al., 2019).

To overcome the above drawbacks, we propose an attention-guided black-box adversarial attack, where a large-scale multiobjective evolutionary algorithm is employed to traverse the salient region of an image. Since the proposed method involves Large-scale Multiobjective Optimization and Attentional mechanism, it is named by LMOA. The main contributions can be summarized as follows:

- **Using the attention mechanism to screen the attacked pixels.** We firstly use the class activation mapping (CAM) and a proxy model to obtain the attention map of the target image. The map strictly limits the attacked pixels so that the perturbations are only allowed to emerge within the salient region. On the one hand, attacking the salient pixels might be more efficient than the entire image(Dong et al., 2020), as these pixels can better reflect the spatial semantic information. On the other hand, screening the pixels is able to reduce the dimensionality of decision variables in the case of high-resolution images, which is beneficial to the black-box optimization.

- **Performing the black-box adversarial attack with a large-scale multiobjective evolutionary algorithm.** We secondly formulate the black-box attack into a large-scale multiobjective optimization problem, in which both attack ability and visual quality of the generated AEs are viewed as two objectives. Then, an optimizer tailored for large-scale optimization is employed. By doing so, a set of Pareto optimal solutions that achieves the balance between two objectives will be obtained, and the final generated perturbations can easily fool the target DNNs while being imperceptible by the human vision.

- **Attacking high-resolution images with high success rate and acceptable visual quality.** Extensive experimental results have been investigated on the ImageNet dataset. The results show that the proposed LMOA can achieve almost 100% success rate of attacks. Compared with the baseline method, LMOA is more competitive to contribute high-resolution AEs with better visual imperceptibility (see Fig. 1).

## 2. Proposed method

The general framework of the proposed LMOA is exhibited in Fig. 2. LMOA mainly consists of two steps: 1) Screening the perturbed pixels with the attention mechanism; 2) Generating the optimal perturbations with a large-scale MOEA.

## 2.1. Screening perturbed pixels with the attention mechanism

By considering the spatial semantic information, LMOA firstly employs the class activation mapping (CAM) (Zhou et al., 2016) to obtain the attention map of the target image. CAM can visualize predicted class scores on any given image, highlighting the region of the object detected by the target DNN. In other words, the obtained attention map reflects the pixels of the interest of DNNs in the classification. Obviously, attacking these pixels that contain the spatial semantic information can fool the target DNN with a relatively higher probability. However, it is tricky to know the gradient information of black-box DNNs, which brings a barrier for using CAM. Therefore, we suggest using a proxy model to obtain an approximated attention map of the input image.

To make the generated adversarial perturbations more effective, only the salient pixels derived from the attention map will be screened as the attacked pixels. To be specific, the attention map of the proxy model is binarized to represent the candidate pixels for attacking. By doing this, the dimension of the decision variables to be optimized is reduced, as the pixels for perturbation is changed. The reduction of the search space also facilitates the convergence of the subsequent MOEA.

## 2.2. Generating perturbations with large-scale MOEA

The black-box attack is firstly formulated as a multiobjective optimization problem below.

$$
\begin{aligned}
\min \quad & f_1 = P(\mathcal{C}(\boldsymbol{I} + \boldsymbol{X}) = \mathcal{C}(\boldsymbol{I})) \\
\min \quad & f_2 = \|\boldsymbol{X}\|_0 \\
\min \quad & f_3 = \|\boldsymbol{X}\|_2 \\
\text{s.t.} \quad & 0 \le u_i + x_i \le 255
\end{aligned} \tag{1}
$$

where $P(\cdot)$ denotes the *confidence probability* of the classification result; $\boldsymbol{I}$ and $\boldsymbol{X}$ represent the *original sample* and *adversarial perturbation*, respectively; $u_i$ is the value of pixel at the $(l, w, c)$ position of $\boldsymbol{I}$, while $x_i$ is the value of perturbation.

As shown in Eq. (1), the proposed multiobjective optimization based black-box attack involves three objective functions. The first one $f_1$ represents the probability that the target classifier $\mathcal{C}(\cdot)$ classifies the generated adversarial example $\boldsymbol{I} + \boldsymbol{X}$ into the correct class $\mathcal{C}(\boldsymbol{I})$. The remaining two functions are both distance metrics, each of which is employed to evaluate the similarity between $\boldsymbol{I} + \boldsymbol{X}$ and $\boldsymbol{I}$. Furthermore, minimizing $l_0$ distance ($l_0$ norm) is to restrict the number of pixels to be attacked, while minimizing $l_2$ distance aims to reduce the change of each pixel. Other than three objective functions, the constraint imposed to $\boldsymbol{X}$ defines the range of perturbation on each pixel based on the intrinsic property of images. Using such a constraint

can effectively reduce the search space and facilitate the convergence of MOEA. Finally, we note that most of the decision variables in $\boldsymbol{X}$ are fixed as zero, and the dimension of perturbations to be optimized is relatively lower than that of $\boldsymbol{I}$. The reason is that the candidate attacked pixels are significantly reduced according to the technique introduced in the last subsection.

Note that, the objective $f_2$ is compatible with $f_3$ in some cases, *e.g.*, a solution with a small objective value on $f_2$ may also has an acceptable performance on $f_3$. Moreover, the value of $f_2$ reflects the sparsity of a solution, that is, minimizing $f_2$ is to find the most sparse adversarial attack. We thereby reformulate the black-box attack below and resort to one of MOEAs tailored for large-scale sparse multiobjective optimization problems (LSMOPs)..

$$
\begin{aligned}
\min \quad & f_1 = P(\mathcal{C}(\boldsymbol{I} + \boldsymbol{X}) = \mathcal{C}(\boldsymbol{I})) \\
\min \quad & f_2 = \|\boldsymbol{X}\|_2 \\
\text{s.t.} \quad & 0 \le u_i + x_i \le 255
\end{aligned} \tag{2}
$$

LSMOPs are characterized as the problems, where the value of most decision variables of their Pareto optimal solutions is zero, and the remaining large-scale variables are one or the other real numbers. Since recent methods can tackle this type of problems (Tian et al., 2020b), we employ the MOEA based on Pareto-optimal subspace learning (MOEA/PSL), which is recently proposed in (Tian et al., 2020a), to solve the problem depicted in Eq. (2).

# 3. Experiments and analysis

In this section, we frst introduce the experimental setup, in cluding the benchmark dataset and parameter setting. Then we compare the proposed algorithm with the work proposed in (Suzuki et al., 2019) in terms of attack ability and visual quality on benchmark dataset.

## 3.1. Experimental setup

The benchmark dataset contains 1000 high-resolution images that are randomly selected from ImageNet-1000 (Deng et al., 2009), which consists of 1000 categories in total. Two DNNs, including the pretrained ResNet-101 (He et al., 2016) and Inception-v3 (Szegedy et al., 2016), are selected as the target models for each compared algorithm. Before the optimization, the resolution of each image is resized according to the input layer of each model, *i.e.,* $224 \times 224 \times 3$ for ResNet-101 and $299 \times 299 \times 3$ for Inception-v3, respectively. All the experiments are carried out on a PC with Intel Core i7-6700K 4.0GHz CPU, 48GB RAM, Windows 10, and Matlab R2018b with PlatEMO (Tian et al., 2017).

*Table 1.* Classification results and corresponding confidences of the original and AEs.

| Image No. | Recognition results and confidence | | | | | | | |
|---|---|---|---|---|---|---|---|---|
| | ResNet-101 | | | | Inception-v3 | | | |
| | $\mathcal{C}(\boldsymbol{I})$ | | $\mathcal{C}(\boldsymbol{I}+\boldsymbol{X})$ | | $\mathcal{C}(\boldsymbol{I})$ | | $\mathcal{C}(\boldsymbol{I}+\boldsymbol{X})$ | |
| $I_1$ | Vulture: | 62.05% | Kite: | 52.37% | Kite: | 15.30% | Kite: | 70.98% |
| $I_2$ | Hotdog: | 95.69% | Cucumber: | 47.46% | Banana: | 10.03% | Banana: | 59.93% |
| $I_3$ | Street sign: | 50.28% | Shopping cart: | 57.47% | Street sign: | 15.19% | Shopping cart: | 88.76% |
| $I_4$ | Wolf spider: | 92.69% | Tarantula: | 51.47% | Wolf spider: | 93.70% | Barn spider: | 43.05% |
| $I_5$ | Knot: | 97.67% | Swab: | 53.19% | Knot: | 92.69% | Swab: | 42.84% |
| $I_6$ | Fig: | 81.77% | Jackfruit: | 22.90% | Fig: | 99.06% | Mushroom: | 44.82% |
| $I_7$ | Hip: | 77.13% | pomegranate: | 42.94% | Hip: | 33.66% | Lenmon: | 33.41% |

*Table 2.* Comparison on the classification accuracy,and average $l_2$ norm of AEs between LMOA and the baseline (Suzuki et al., 2019).

| Target model | Attack method | Classification accuracy (%) | Avg. $l_2$ norm |
|---|---|---|---|
| ResNet-101 | N/A | 77.10 | N/A |
| | The baseline | 9.30 | 73949.58 |
| | LMOA | 0.00 | 967.51 |
| Inception-v3 | N/A | 78.40 | N/A |
| | The baseline | 15.60 | 152008.99 |
| | LMOA | 0.30 | 1186.34 |

## 3.2. Results and analysis

Table 1 shows the classification results and confidence probabilities of seven randomly selected original images and their corresponding AEs obtained with two DNNs. From the table, two remarks can be concluded as follows. Firstly, for the images that are correctly classified by the two DNNs, the proposed LMOA finally generates AEs that fool the models with high confidence probabilities (at least 22.9%, most of the results over 40%). Secondly, for the images that are misclassified by DNNs ($I_1$ and $I_2$ against Inception-v3), the proposed algorithm improves the confidence probabilities of the incorrect label (from 15.3% to 70.98% on $I_1$, 10.03% to 59.93% on $I_2$).

Table 2 compares LMOA and another MOEA-based black-box attack method (Suzuki et al., 2019), which adopts block-division method and formulates a MOP solved by MOEA/D (Zhang & Li, 2007). From the table, we can observe that 77.1% of the benchmark images can be correctly classified by ResNet-101, while 21.6% of them are misclassified by Inception-v3. After performing the attack with the baseline [35], only 9.3% and 15.6% of the images are correctly classified by the two DNNs, respectively. By contrast, LMOA has successfully attacked most of the images and fooled both of the two models. More concretely, LMOA has achieved a 100% success rate on ResNet-101. We also notice that LMOA performs better in terms of average $l_2$ norm between the original image and the generated AE. Fig. 3 also visual-

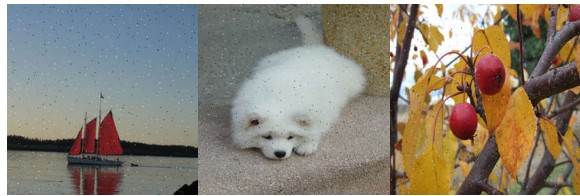

(a) AEs generated by the baseline method

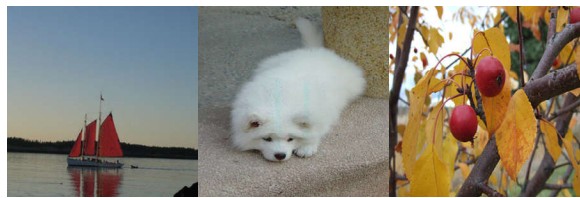

(b) AEs generated by the proposed LMOA

*Figure 3.* The visual comparison of the AEs generated by the baseline method and the proposed LMOA.

ized attack results obtained by the two methods. From the figure, we show that the baseline method adds some visible noises to each entire image, which can be easily captured by the human vision. For LMOA, the perturbations are much more difficult to be perceived compared with the baseline method. The advantage is attributed to the usage of the attention mechanism and large-scale MOEA.

## 4. Conclusions

In this paper, we propose a novel attention-guided black-box adversarial attack, where the adversarial perturbations are only added to several pixels in the salient region. Besides, MOEA/PSL is used to search for the optimal perturbation, and the algorithm is used to solve MOPs, where the Pareto optimal solutions are sparse. Experimental results show that the proposed LMOA can perform effective black-box attacks against high-resolution images (with nearly 100% success rate and high visual quality). Moreover, comparing with the baseline method, the proposed LMOA is more suitable to high-resolution images from ImageNet dataset.

## Acknowledgments

This research work is partly supported by National Natural Science Foundation of China No.61872003, No.61502009, State Key Laboratory of Computer Architecture (ICT,CAS) under Grant No. CARCHB202018.

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
