# OpenReview forum: "Attention-Guided Black-box Adversarial Attacks with Large-Scale Multiobjective Evolutionary Optimization"
_ICML.cc/2021/Workshop/AML — ICML 2021 Workshop AML Poster_

### Official Review · Reviewer_UyDy · 2021-06-20
**Using the class activation mapping to improve the quality of generating adversarial examples using evolutionary optimization for ImageNet models**

**Rating:** Accept
**Confidence:** 3

**Review:**

The authors propose to only apply adversarial perturbations to the more important area of the image. They formulate the black-box attack problem as a multiobjective optimization problem and use MOEA/PSL to solve it. The adversarial examples generated by the proposed method have good visual quality as well as smaller average $\ell_2$ norm.

The paper is clear.

Pros:
1. This work can trade-off between multiple objective when generating adversarial examples, e.g., both $\ell_0$-norm and $\ell_2$-norm perturbation size are considered in this work. Existing attacks usually consider only one distance measurement.
2. The generated adversarial examples have better visual quality than another baseline evolutionary-optimization-based method.

Cons:
1. Other decision-based black-box adversarial attacks are not included in the related work and the experiments, e.g., Sign-OPT, and HSJA.
2. For black-box attacks, we also care about their query efficiency, since under real-world black-box scenarios, the number of queries to the target model is usually quite limited. Providing the number of queries to the target model in the experiments would make the evaluation more comprehensive.

---

### Decision · Program_Chairs · 2021-06-21

**Decision:**

Accept (Poster)

**Comment:**

This paper proposed to apply adversarial perturbations to more important area of the image. The problem is well formulated and MOEA/PSL is adopt to solve it. The generated adversarial examples have good visual quality.